# A Nonlinear Probabilistic Pitting Corrosion Model of Ni–Ti Alloy Immersed in Shallow Seawater

**DOI:** 10.3390/mi13071031

**Published:** 2022-06-29

**Authors:** Špiro Ivošević, Gyöngyi Vastag, Nataša Kovač, Peter Majerič, Rebeka Rudolf

**Affiliations:** 1Faculty of Maritime Studies Kotor, University of Montenegro, Put I Bokeljske Brigade 44, 85330 Kotor, Montenegro; 2Faculty of Sciences, University of Novi Sad, Trg Dositeja Obradovića 3, 21000 Novi Sad, Serbia; djendji.vastag@dh.uns.ac.rs; 3Faculty of Applied Sciences, University of Donja Gorica, Oktoih 1, 81000 Podgorica, Montenegro; natasa.kovac@udg.edu.me; 4Faculty of Mechanical Engineering, University of Maribor, Smetanova Ulica 17, 2000 Maribor, Slovenia; peter.majeric@um.si (P.M.); rebeka.rudolf@um.si (R.R.)

**Keywords:** Ni–Ti alloy, pitting corrosion, seawater, EDX analysis, nonlinear, probabilistic

## Abstract

The degradation of metal materials in a marine environment represents the consequence of the electrochemical corrosion of metals under the influence of the environment. The application of new materials in the maritime industry requires experimental, real-world research on the form of corrosive damage and the intensity of the corrosion. This paper analyses the pitting corrosion of a rod-shaped nickel–titanium (Ni–Ti) alloy that was produced by means of the continuous casting method. In total, three samples were posted in a real seawater environment and analysed after 6, 12, and 18 months. Pits were detected on the Ni–Ti alloy after 18 months of exposure to the marine environment. The database on pitting corrosion was created by measuring depth in mm, which was performed by means of a nonlinear method, and by the generation of an artificial database of a total of 120, gauged in critical pit areas. The data were obtained by the application of a nonlinear model, and under the assumption that corrosion starts after 12 months of exposure in the corrosive marine environment. The EDX analysis of the Ni–Ti alloy composition inside the pits and on the edges of the pits indicated that the corrosion process in the hole of the pit occurs due to the degradation of the Ni.

## 1. Introduction

Corrosion is well defined as the destructive attack on a material by a reaction with its environment [1,2,3,4]. As a result of the corrosion process in different environments, the basic materials lose some of the mechanical properties of the structure, such as strength, ductility, and impact strength. The impact of corrosion depends on the industry, and can have a major economic significance. The World Corrosion Organization (WCO) estimates that the annual cost of corrosion worldwide is around USD 2.4 trillion (3% of the world’s GDP) [5].

Between different industrial fields, the corrosion process of marine environments has been studied for many years, due to the different structural deterioration and product loss in different structural elements of vessels, the offshore and oil industry, or marine renewable energy sources (e.g., tidal energy, wave energy, ocean thermal gradient, salinity gradient, offshore wind, and biomass energy) [6].

Various physical forms of corrosion, such as general, intergranular, pitting, galvanic, crevice, stress, cavitation corrosion, corrosive fatigue, etc., can be found to date [7]. Between the different physical forms of corrosion, previous studies were focused mainly on the development of models of general corrosion and pitting corrosion.

Pitting corrosion is defined as localised corrosion of a metal surface, confined to a point or small area that takes the form of cavities [8]. The total loss of the metal structure may be very small, but the local rate of corrosion attack can result in holes in the structure, which can lead to early catastrophic failure (pollution). Development of a pit attack can be described in four stages: passive film breakdown, pit initiation, metastable pitting, and pit propagation [9].

In order to understand the pitting corrosion of steel structures, there are many techniques that can identify the presence of pitting [10,11], such as visual inspection, non-destructive testing (NDT), surface analysis techniques, and probabilistic approaches for pit identification. All of these techniques can be used to describe structural failure due to pitting, and to describe the characteristics of pits, such as pit depth, pit rate, or pit density. Based on the reviewed literature it is obvious that problems connected with pitting corrosion are stochastic and probabilistic, and require interdisciplinary concepts including surface science, metallurgic science, materials science, hydrodynamics, and chemistry. There are a lot of different corrosion models that describe pitting corrosion. Many authors describe the corrosion process as depending on different environmental factors and structural elements attacked by pitting corrosion. Using regression analysis, Katano et al. proposed a predictive model for pit growth where pitting depth was a dependent variable and environmental factors appeared as independent variables [12]. Melchers considered the effect of dissolved oxygen and the effect of water velocity on pitting depth [13]; he confirmed that depth is not an independent variable in marine immersion corrosion of mild steel [14,15].

Guedes Scares et al. proposed a corrosion wastage model based on a nonlinear time-dependent function that described the effects caused by different environmental factors (i.e., salinity, temperature, dissolved oxygen, pH, and flow velocity) on the pitting corrosion behaviour of steel plates totally immersed in saltwater conditions [16]. Melchers studied the statistical characteristics of pitting corrosion, represented by the extreme “Gumbel” value distribution [3]. Furthermore, he recommended using the normal distributions to represent the extreme pit depth of super-stable pitting [13]. Khan et al. [17] recommended the use of extreme value statistics in pitting corrosion to investigate extreme values, and proposed linear, power-law, and logarithmic extreme value models [17,18]. Mohammad et al. (2012) [19] proposed a prediction model of pitting corrosion characteristics using artificial neural networks (ANNs), while Valor et al. (2013) [20] proposed Markov chain models for the stochastic modelling of pitting corrosion [20,21,22], and Velazquez et al. (2014) [5] described the pit initiation as a non-homogeneous Poisson process (NHPP). 

The development of the maritime industry in the 21st century requires the use of smart materials, as they are shape-memory materials. The term ‘‘Shape-Memory Effect’’ was recognised and described by William Buehler and Frederick Wang in a Ni–Ti alloy (nitinol) in 1962 [23,24]. To date, nitinol has been used in the medical, transport, and robotics industries, as well as different civil structures [25]. 

Shape-memory materials based on Ni–Ti can be produced with appreciable thermomechanical properties using different production techniques, such as conventional casting or powder metallurgy processes. Conventional casting can be vacuum arc remelting (VAR) or vacuum induction melting (VIM). Powder metallurgy production can be by conventional methods (e.g., conventional sintering, self-propagating high-temperature synthesis, hot isostatic pressing, metal injection moulding, spark plasma sintering) or advanced manufacturing (e.g., selective laser melting, selective laser sintering, laser-engineered net shaping, electron beam melting) [26].

The subject of this research is the testing of a continuously cast Ni–Ti alloy in the shape of a rod. Despite previous research [27,28,29], when the pitting corrosion rate was analysed in this paper using a linear model (assuming the onset of corrosion after 0, 6, and 12 months) we obtained a database from which we developed a nonlinear corrosion model (assuming that the corrosion process started after 12 months of exposure). Furthermore, using the EDX method, and analysing the changes in the chemical composition of the Ni–Ti alloy at the edge and in the depth of the crater of the located pit, the corrosion process of Ni–Ti pitting following immersion in seawater can be described.

## 2. Materials and Methods

### 2.1. Preparation of the Ni–Ti Alloy

The Ni–Ti alloy in the shape of a rod was produced via vacuum remelting (furnace Leybold Heraeus, Finland) and continuous casting processes with a laboratory-scale vertical continuous casting device (Technica Guss, Würzburg, Germany). The furnace was connected to a power source of 60 kW and produced a medium frequency (4 kHz) that enabled the remelting and casting of the alloy. Pure components were used for preparation of the Ni–Ti alloy: Ni (99.99 wt.%) and Ti (99.99 wt.%), delivered by Zlatarna Celje d.o.o. (Zlatarna Celje d.o.o., Celje, Slovenia). The Ni:Ti weight ratio was 2:1. The Ni–Ti alloy was cast into a rod with ϕ = 12 mm. To perform the experimental testing, samples with dimensions of 2r = 12 mm and l = 50 mm were prepared with electro-erosion cutting. The samples had an incision with a thinner diameter, so that we could tie a flex to them and, thus, locate the samples at the desired location in the sea. All Ni–Ti alloy samples were ground on the surface after electro-erosion to remove contamination, thus enabling the performance of corrosion tests as realistically as possible. In total, three test Ni–Ti samples were deposited in seawater and analysed after 6, 12, and 18 months of exposure. 

Before testing in real seawater, the chemical composition of the test samples was measured by inductively coupled plasma (ICP) and X-ray fluorescence (XRF), in order to check the chemical composition of the Ni–Ti alloy samples. 

### 2.2. Proposed Problem and Related Methodology

Motivated by previous research [29,30,31,32], in this paper we analysed the pitting corrosion of a Ni–Ti alloy in the marine environment. Specifically, 3 test samples of Ni–Ti alloy were immersed in the sea, near the coast, at a depth of three metres, during the period from September 2018 to March 2020. 

In a previous study, a linear corrosion model of pitting formed over an alloy exposed to immersion in seawater for 18 months was analysed, varying the assumption of an initial corrosion onset period of 0, 6, and 12 months [29]. As no pitting corrosion was observed on the samples exposed to the influence of the sea after 6 and 12 months, this paper analyses the rapid corrosion process, assuming that pitting occurs after 12 months of exposure.

During the research between August 2018 and March 2020, the parameters of the seawater (i.e., temperature and salinity) were measured by the Institute of Biology at the University of Montenegro. The minimum seawater temperature was in January 2019 (11.7 °C), while the maximum temperature was in August 2020 (25.90 °C). Considering salinity at the measurement location, the minimum salinity was in December 2019 (14.00‰), while the maximum salinity was in August 2019 (39.00‰). Some additional information is part of the previous research [30].

Considering the above, the conceptual model of the research is shown in Figure 1. The research was conducted in two ways: In the first, the nonlinear pitting corrosion model was analysed on the basis of SEM measurements and artificial data collected on the basis of the extreme pitting depth, expressed in millimetres. In the second direction, based on EDX analyses, the values of the chemical composition of the pitting edge and hole were collected, in order to analyse changes to the Ni–Ti alloy surface.

#### 2.2.1. SEM Observation

The SEM imaging was performed with a Quanta 200 3D (FEI, Hillsboro, OR, USA) scanning electron microscope using the Everhart–Thornley Detector for secondary electrons, with an acceleration voltage of 10 kV, in high-vacuum mode. For imaging, a magnification of 50× with a working distance of around 29 mm was used, in order to maximise the depth of field in the image. In order to reconstruct a 3D model from the images using stereophotogrammetry, and to obtain pitting depth information calculated by trigonometry, the sample must be viewed from several angles around the Z-axis [33]. The images of the samples were taken at tilt angles from 10° to 50° with 5° steps. The software MountainsSEM Expert (MountainsSEM Expert version 8.2.9564, Digital Surf, Besançon, France), was used for the 3D model reconstructions and pitting depth measurements. The pitting depth measurements were taken from the peak and valley positions of 6-line topographical profiles at various locations on the pitting hole.

#### 2.2.2. EDX Analysis

During the 18 months of exposure in the seawater environment, pitting was located on the Ni–Ti sample, as shown in Figure 2.

The chemical composition of the selected Ni–Ti alloy surfaces was determined through the use of a high-resolution field emission SEM (Sirion 400 NC; FEI, Hillsboro, OR, USA), equipped with an INCA 350 EDX detector (Oxford Instruments, Abingdon, Oxfordshire, UK). The EDX semi-quantitative analysis determined the chemical composition of the Ni–Ti alloy surface after corrosion, as well as the content of the elements on the surface edge and hole of the pitting. Sixty spectra were considered in total. Five analysis locations were in the hole of the pitting and five were on top of the edge. The chemical composition of the metal surfaces was scanned for each analysis location—up to six spectra per location under the magnification of 200 μm and 300 μm.

## 3. Results

Chemical analysis of the percentages of Ni and Ti of the initial test samples (before being exposed to the seawater environment) is shown in Table 1 [32].

### 3.1. Development of the Ni–Ti Alloy Nonlinear Probabilistic Pitting Model

The pitting depth measurements on the selected six-line topographical profiles on the pitting hole are shown in Figure 3.

Based on the performed SEM measurements, an artificial enlargement of the input database was performed, testifying to the depth of corrosion. Specifically, on each image A–F from Figure 3, a total of 20 measured values was generated in such a way that, on a scale from 1 to 5 mm, the value of corrosion depth was selected and measured, as shown in Figure 4. Out of a maximum of 50 measured data, 20 data were selected corresponding to the critical shape of the pitting crater and the measured pitting depth values at a length of two millimetres. In this way, a total of 20 measured values was generated for each image from A–F, i.e., a total of 120 measured values. The minimum, maximum, and average values for each section from A–F are presented in Table 2.

In order to summarise the sample of derived pitting corrosion values in a straightforward manner, the most common descriptive statistics are given in Table 3. Typical measures of central tendency (i.e., mean value), with values of variability (i.e., variance, standard deviation, coefficient of variation, standard error) and sample shape (i.e., skewness, excess kurtosis) [34], are presented in the first column of Table 2. The calculated values of these statistics are given in the second column of Table 2. The next two columns show the percentiles of the sample.

The corrosion loss of the Ni–Ti alloy exposed to the environment can be expressed as a function of the strength of time [35,36]. Depending on the length of exposure of metal structures to environmental influences, the observed time intervals are usually expressed in months or years. Paik and Thayamball [36] proposed a corrosive model that takes into account a time parameter that affects the delay of the onset of corrosive processes, and is usually described as the effectiveness of a protective coating. This model can be displayed in the following format:(1)$$d\left(t\right)=\left\{\begin{array}{cc} c_{1}{\left(t-T_{cl}\right)}^{c_{2}}, & t>T_{cl}\\ 0, & otherwise \end{array}\right.$$
where the corrosion depth is denoted by $d\left(t\right)$, the time of exposure to the environment is denoted by t, while $T_{cl}$ is the time when the corrosive processes begin to develop. Parameters $c_{1}$ and $c_{2}$ are unknown parameters that take positive real values, and are determined in the process of fitting the model to the empirical data. The coefficient $c_{1}$ represents the corrosion rate, and is usually measured in mm/year or nm/month. Parameter $c_{2}$ regulates the intensity of the time component of the model. Values of $c_{2}=1$ or $c_{2}=1/3$ have been suggested in the literature. Preliminary results of testing this model on empirical data for a Ni–Ti alloy show that pitting corrosion did not occur until the 12th month of exposure of the samples to environmental influences, so it is suitable for $T_{cl}$ to be 12. However, the best value for the parameter $c_{2}$ is slightly higher than the suggested 1. Tests showed that, in the case of a Ni–Ti alloy, the best fit of Model (1) is obtained if $c_{2}=1.1$.

Bearing these preliminary results in mind, in the continuation of the statistical analysis, a nonlinear model (Model (1)) is used, which takes the following form after the inclusion of the experimentally determined parameters $T_{cl}=12$ and $c_{2}=1.1$:(2)$$d\left(t\right)=\left\{\begin{array}{cc} c_{1}{\left(t-12\right)}^{1.1}, & t>12\\ 0, & otherwise \end{array}\right.$$

Model (2) is used to describe the depth of pitting corrosion formed on the Ni–Ti alloy samples. In the statistical analysis aimed at determining the value of the parameter $c_{1}$, the database described in Table 2 was used.

Corrosive processes can be viewed as deterministic and stochastic. Corrosion is a process that depends on a large number of factors, and is subject to uncertainty. Therefore, in this paper, the corrosion rate is treated as a stochastic continuous variable that depends on the time of exposure of the samples to environmental influences. By expressing the coefficient $c_{1}$ from the formed Model (2), Model (3) is obtained, suitable for fitting continuous distributions into the calculated values for corrosion rate:(3)$$c_{1}=\frac{d\left(t\right)}{{\left(t-12\right)}^{1.1}},\;t>12$$

Continuous distributions are described with the probability density function (PDF) and cumulative density function (CDF). Common labels for PDF and CDF are $f\left(x\right)$ and $F\left(x\right)$, respectively. The best fitted normal distribution was obtained using 120 values for pitting corrosion depth and applying Model (3). Normal distribution was selected as a candidate for adequate continuous distribution, based on recommendations from the scientific literature dealing with pitting corrosive processes [13]. The optimal values of the normal distribution parameter were determined using the maximum likelihood estimation method [37]. The normal distribution thus formed is characterised by the mean value μ=0.069 and standard deviation σ=0.041. More precisely, the best fitted normal distribution describing the $c_{1}$ (corrosion rate) parameter has PDF and CDF represented by Formulae (4) and (5), respectively:(4)$$f\left(x\right)=9.7303e^{-297.442{\left(x-0.069\right)}^{2}}\;\to \mathrm{for}-\infty <x<+\infty$$
(5)$$F\left(x\right)=9.7303\int_{-\infty }^{x}e^{-297.442{\left(t-0.069\right)}^{2}}dt$$

It is worth noting that the obtained value of the parameter μ=0.069 is in accordance with the mean value of the empirical data shown in Table 3. Knowing the formula for CDF [38], the probability of occurrence of pitting corrosion depth values for a Ni–Ti alloy can be estimated as follows:(6)$$P\left(r_{1}{\left(t-12\right)}^{1.1}\leq d\left(t\right)\leq r_{2}{\left(t-12\right)}^{1.1}\right)=P\left(r_{1}\leq c_{1}\leq r_{2}\right)\approx F\left(r_{2}\right)-F\left(r_{1}\right),$$
where t>0 denotes the elapsed time expressed in months from the beginning of the exposure of the Ni–Ti alloy to the environmental influences.

The fitted model and the derived expression shown by Formula (6) can be used to predict the value of corrosion depth after a given time of exposure of the Ni–Ti alloy sample to the seawater environment. For example, if the alloy is released into the seawater environment for 15 months (i.e., t=15), and if the corrosion rate is lower than 0.164 mm/month, we come to the following conclusion:(7)$$P\left(c_{1}\leq 0.164\right)\approx F\left(0.164\right)\approx 0.99=P\left(d\left(15\right)\leq 0.55\right).$$

Thus, after 15 months of exposure of the Ni–Ti alloy to the influence of the marine environment, with a probability of 0.99, it can be stated that the corrosion depth will reach a value of 0.55.

A graph of the PDF of the best normal distribution function that describes the pitting corrosion data of the Ni–Ti alloy adequately (represented by Formula (4)) is shown in Figure 5. This figure additionally shows the empirical data of pitting corrosion grouped in bins in the form of a histogram.

The Kolmogorov–Smirnov (KS) test was applied, with the aim of determining the goodness of fit of the previously formed normal distribution [39,40]. To determine how effectively normal distribution monitors empirical data, the following hypotheses were set:

**Hypothesis** **H0**:*Data follow the normal (0.069, 0.041) distribution*;

**Hypothesis** **Ha**:*Data do not follow the stated normal distribution*.

Standard values of significance levels α (i.e., 0.2, 0.1, 0.05, 0.02, and 0.01) were observed, and a corresponding critical value was calculated for each value. By comparing the calculated values of the test statistics with the critical values determined in this way, it is possible to determine whether hypothesis H0 is rejected or not. In addition, the KS test accepts the null hypothesis for all test statistic values less than the calculated *p*-values. As can be seen in Table 4, the obtained value of the KS test statistic was less than the critical values determined for all selected significance levels. Additionally, the KS test statistic was lower than the *p*-value, indicating that H0 could not be rejected.

This section may be divided by subheadings. It should provide a concise and precise description of the experimental results, their interpretation, and the experimental conclusions that can be drawn.

### 3.2. Ni–Ti Alloy Pitting Process

After the immersion of the Ni–Ti samples in seawater, SEM and EDX analyses were used to detect the onset of pitting corrosion and the pit formation mechanism.

The EDX analyses detected several elements (C, Na, Mg, Al, Si, S, P, Cl, K, Ca, and Fe) that were considered the remains of substances from the sea (Table 5 represents spectral data for Figure 6b). The contents of Ti and Ni in the actual surface varied on the edges of the pits and inside the pits. The weight percentage of Ti on the pit edges did not differ significantly from that of the Ti detected inside the pits. On the other hand, the weight percentage of Ni was considerably lower inside the pits than on the edges, indicating the depletion in Ni content due to corrosion and material diminution. In most of the analysed spectrums for the inner parts of the pits, Ni was not even detected, which particularly lowered the mean value of Ni in 30 of the examined spectra. Table 6 shows the Ti and Ni contents (in wt.%) of the analysed spectrums without the aforementioned elements from the sea. Depending on the surface coverage of these elements, the detected Ti and Ni contents can be as low as a few percent, while the other elements—not shown here—take up the rest of the content in the given analysis site.

The EDX analysis’ lower detection limit is considered to be 0.1 wt.%. For major constituents with a mass content greater than 10 wt.%, there is a relative uncertainty of ±2% [41]. For minor constituents with a mass content lower than 10 wt.%, the relative uncertainty is considered to be up to 50% for standardless analyses [42]. As the elemental wt.% between samples varied significantly more than the described analysis errors, the obtained values were considered relevant for the examination of these samples after their exposure to the seawater environment.

Figure 7a–c show the surface of the Ni–Ti alloy after exposure to seawater for 6 months, 12 months, and 18 months.

Table 7 shows the average chemical composition of the Ni–Ti surface based on the EDX analyses after varied lengths of exposure.

The identification of the time of the cracking point of the passive layer under the influence of aggressive (chloride) ions, as well as the prediction of the pit changes over time, plays an important role in the practical application of the Ni–Ti alloy. Therefore, an additional EDX analysis of the examined Ni–Ti alloy determined the chemical composition of particular pits whose depth varied between ~0.3 and 1.0 mm. The analysis was conducted separately at five points on the pit edges and five points inside the pits. The chemical composition of the metal surface was scanned for each analysed point with up to six spectra, under the magnification of 200 μm and 300 μm. Figure 7 shows the locations of the pits subjected to the EDX analysis.

Figure 8, Figure 9 and Figure 10 exhibit the comparison of the content of particular chemical elements that were detected by the EDX analysis on the pit edges and inside the pits. Figure 8 shows the content of Ni in the pits and on the edge of the pits.

Figure 9a shows the content of Ti (a) and oxygen in the pits and on the edge of the pits.

Figure 10a shows the content of chlorides (a) and the total content of inorganic cations in the pits and on the edges of the pits.

## 4. Discussion

The conducted statistical analysis was motivated by three facts known in the literature:Corrosive processes can be modelled as a power function of environmental exposure times;Corrosive processes are stochastic in nature or, more precisely, the pitting corrosion rate can be viewed as a continuous random variable;The normal distribution is suitable for representing the extreme depth of the pitting corrosion process of a super-stable alloy.

By applying these assumptions over a database of 120 measurements of pitting corrosion depth of a Ni–Ti alloy, a nonlinear probabilistic model for pitting corrosion rate was formed in the process of fitting the theoretical (normal) distribution. Statistical analysis showed that the pitting corrosion rate can be modelled very successfully as a continuous random variable that behaves in accordance with the characteristics of the normal distribution.

The implemented statistical methodology for pitting corrosion modelling could be applied to other continuous distributions, as well as to other types of alloys on which pitting corrosion was detected. In addition, it is evident that the extreme values of pitting show certain laws of behaviour, indicating the possibility of testing the extreme value theory in describing the values of corrosive processes.

As shown in Figure 6a and Table 7, the surface of the Ni–Ti alloy that was exposed to seawater for 6 months was covered unevenly with inorganic salts from the sea (Na, Mg, Ca, K, Cl, Si, …, ${\mathrm{SO}}_{4}^{2-}$) and corrosion products (Ti and Ni oxides). The inhomogeneous layer on the surface cannot ensure protection from corrosion [43,44], as confirmed by the EDX analysis conducted after 12 months of the exposure of the alloy to seawater. The analysis detected excessive corrosion on the alloy and a notable reduction in the contents of Ti and Ni (Table 7). A low amount of chloride ions was also detected on the alloy’s surface by means of the EDX analysis after 12 months of exposure (Cl < 0.8 wt.%).

Figure 6b shows that after 12 months of exposure to seawater, the surface of the analysed Ni–Ti alloy was more homogeneous and contained fewer inorganic salts (Ca-silicate) than after 6 months of exposure. The more homogeneous surface of the alloy was the result of the formation of a uniform layer of oxides on the surface of the examined alloy [45,46,47]. In that regard, previous studies indicated that the corrosion of Ni–Ti alloys begins on surface defects that emerge after damage to the original oxide layer. During the corrosion process Ni is usually released from an alloy, whereby the remaining Ti reacts with dissolved oxygen from the aqueous solution and creates Ti oxides around the surface defects [45,46]. This layer contains mainly Ti oxides in the outer layer and Ni–Ti in the inner layer [48,49]. However, the oxide layer formed on the Ni–Ti alloy surface always contains a certain amount of Ni that cannot be released in case of corrosion [50,51]. If the formed oxide layer covers the entire surface of the Ni–Ti alloy, corrosion is inhibited temporarily, because the oxide layer is resistant to chloride ions [47,52]. In the case of our examined alloy, the contents of Ni and Ti did not decrease during the prolonged exposure (Table 7), confirming the statement that the formed layer prevents further corrosion. The length of exposure and the increase in the temperature of the corrosive solution notably decreased the passivation ability of the Ni–Ti alloy [53].

On the other hand, the EDX analysis conducted after 12 months of exposure to seawater detected a considerable content of carbon on the Ni–Ti alloy’s surface (Table 7) which, according to previous studies, is the consequence of the presence of algae, mould, and other organic marine deposits on the surface of the Ni–Ti alloy [54]. The presence of the marine organisms was not surprising, considering the fact that the research was conducted in the spring and summer (March—August 2020), when the activities of the marine organisms are more prominent due to higher sea temperatures.

The amount of chloride ions detected on the Ni–Ti alloy surface after 12 months of exposure was significantly higher (Cl~3 wt.%) than after the first 6 months, while pitting corrosion was not detected on any of the analysed samples.

A thick layer of corrosion products (Figure 7c) with increased oxide content and decreased carbon content (Table 7) was observed on the surface of the analysed Ni–Ti alloy after 18 months of exposure to seawater. Based on Table 7, the amounts of Ni and Ti remained the same as they were after 12 months of exposure, substantiating the claim that the formed layer successfully prevented corrosion. It should also be noted that the amount of chloride ions on the alloy surface was reduced (Table 7), while the appearance of damage in the form of pits increased, as shown in Figure 7c. This means that the additional 6 months of exposure to seawater led to the emergence of pitting corrosion on the surface of the examined Ni–Ti alloy.

The obtained results confirm the findings of previous studies, which claim that the extended exposure of alloys to a corrosive environment amplifies the oxide layer on the surface of Ni–Ti alloys [45]. The amplified oxide layer is a consequence of the migration of oxide ions through oxide anion vacancies. In addition to oxide anions, other anions found in the corrosive environment can also migrate through the formed oxide layer [55]. The migration of chloride ions is particularly important, as the concentration of chloride ions is usually high—just below the concentration of oxide ions. The accumulation of chloride ions was observed on the alloy surface after 12 months. After migration, the accumulation of chloride ions between the metal surface and the formed oxide layer leads to the cracking of the passive oxide layer, thus leaving the surface unprotected and susceptible to pitting corrosion [55,56].

Hu et al. [45] described the mechanism of the formation of pits on the surface of alloys. The amount of Ni registered on the pit edges and inside the pits (Figure 8) confirmed that the pits on the surface of the Ni–Ti alloy that was exposed to seawater for 18 months appeared in accordance with the mechanism described by Hu et al. Specifically, at all points measured on the pit edges, the registered amount of nickel was changeable, while inside the pits (regardless of the pit depth) nickel was almost non-existent (with the exception of one point). The difference in chemical composition inside the pits and on the pit edges indicates that the alloy corrodes through the initial release of nickel ions into the environment, while the remaining Ti oxides enrich the alloy and form an oxide layer. This finding is substantiated by the registered amounts of Ti and oxygen in the pits and on the pit edges (Figure 9a,b). As illustrated in Figure 9a, there was not a significant difference in the content of Ti on the pit edges and inside the pits, meaning that the Ti content does not vary significantly during the formation of pits (Ti remains on the surface of the alloy during corrosion).

Figure 9b shows that, regardless of pit depth, the oxygen content measured at all points inside the pits had approximately the same value and was, on average, slightly higher than the oxygen content measured on the pit edges. This means that the bottom of the pits could be covered with an oxygen layer. Similarly, T. Hu et al. stated that the pits formed during corrosion are possibly blocked by the Ti oxides formed inside the pits [45,46,47]. 

The increased amount of chloride ions could be a consequence of the deposition of inorganic salts on the bottom of the pits. However, this assumption was disputed by the fact that the total content of inorganic cations that can form chlorides (Ca, Mg, Na) (Figure 10b) was not notably higher inside the pits than on the edges. The increased content of chloride ions was therefore not the consequence of the accumulated inorganic salts on the bottom of the pits. As Hu et al. explained, chloride ions become incorporated into the passive layer of Ti oxides over time, and cause the deepening of the pits.

## 5. Conclusions

From the research of a nonlinear probabilistic pitting corrosion model of Ni–Ti alloy immersed in shallow seawater, the following can be concluded:The appearance of damage in the form of pitting on the Ni–Ti surface was first registered after an exposure time of 18 months.The difference in chemical composition inside the pits and on the pit edges indicates that the Ni–Ti alloy corrodes through the initial release of Ni ions into the environment, while the remaining Ti oxides enrich the alloy and form an oxide layer.Chloride ions become incorporated into the passive layer of Ti oxides over time, and cause the deepening of the pits.The extreme value of pitting can be described as a continuous random variable with the characteristics of normal distribution.

## Figures and Tables

**Figure 1 micromachines-13-01031-f001:**
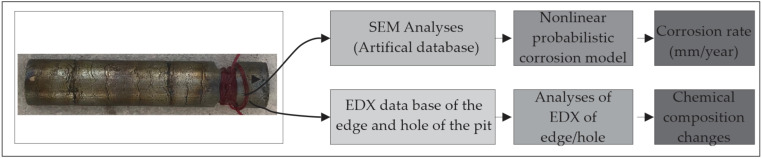
The scheme of the conceptual model of the research.

**Figure 2 micromachines-13-01031-f002:**
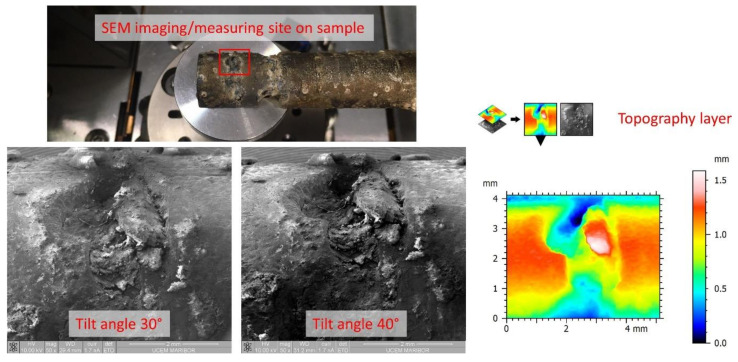
Scheme of SEM imaging and 3D model reconstruction of the sample for depth measurements.

**Figure 3 micromachines-13-01031-f003:**
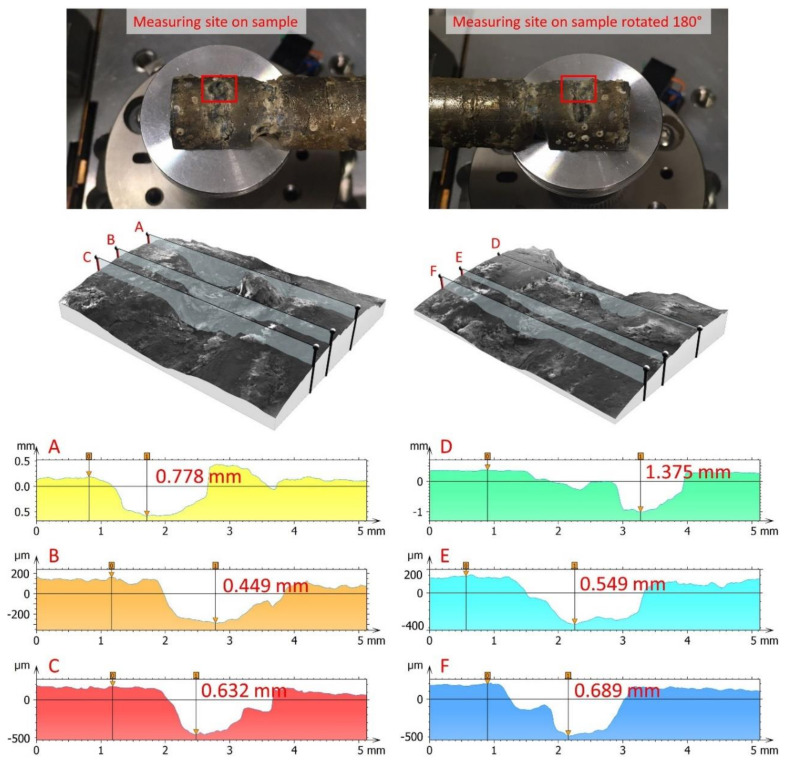
Pitting depth measurements from the line profiles of the pitting holes on the Ni–Ti samples, produced with 3D models reconstructed from the SEM images. The depth was measured from the difference between the peak and valley positions of the line profiles. (**A**–**F**) topographical line profile at various location on the pitting hole.

**Figure 4 micromachines-13-01031-f004:**
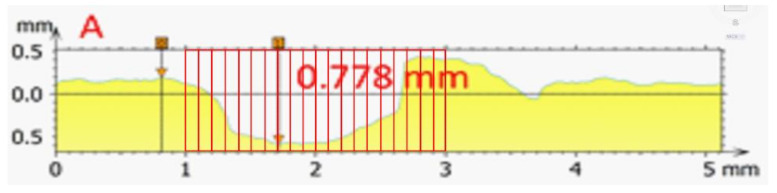
Generation of 20 pitting depth measurements on the Ni–Ti surface in the critical part of the pitting hole.

**Figure 5 micromachines-13-01031-f005:**
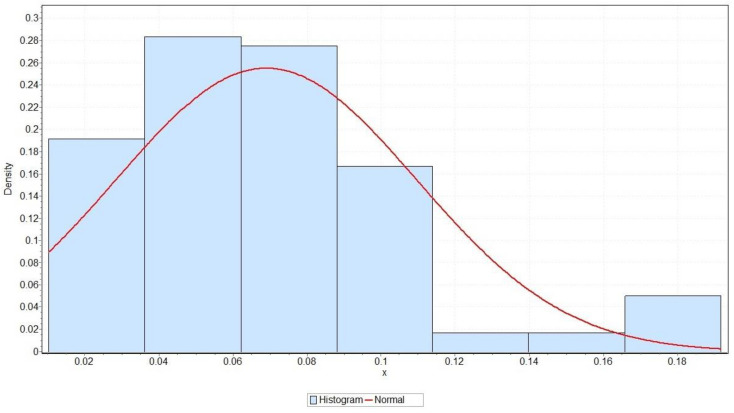
Probability density function (PDF) graphics for the best fitted normal (0.069, 0.041) distribution for the Ni–Ti alloy’s pitting corrosion rate.

**Figure 6 micromachines-13-01031-f006:**
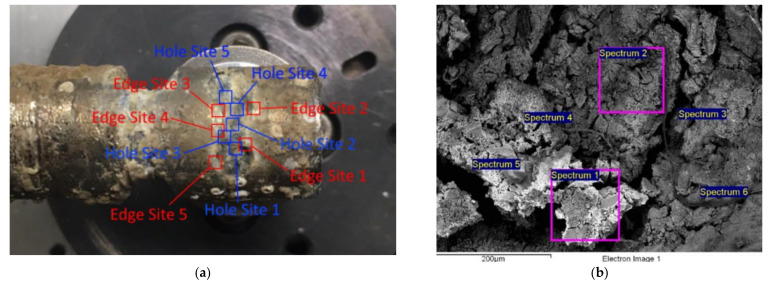
The Ni–Ti alloys after 18 months of exposure to the seawater environment: (**a**) the surface points subjected to the EDX analysis of pitting corrosion, (**b**) the SEM image with selected points for the EDX analyses.

**Figure 7 micromachines-13-01031-f007:**
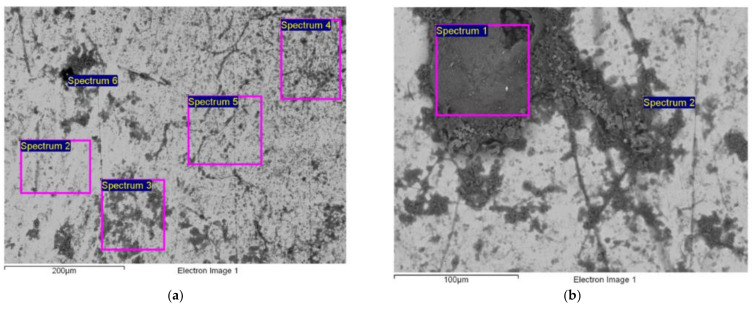
The characteristic microstructure of the surface of the Ni–Ti SMA after exposure to seawater for (**a**) 6 months (**b**) 12 months, and (**c**) 18 months.

**Figure 8 micromachines-13-01031-f008:**
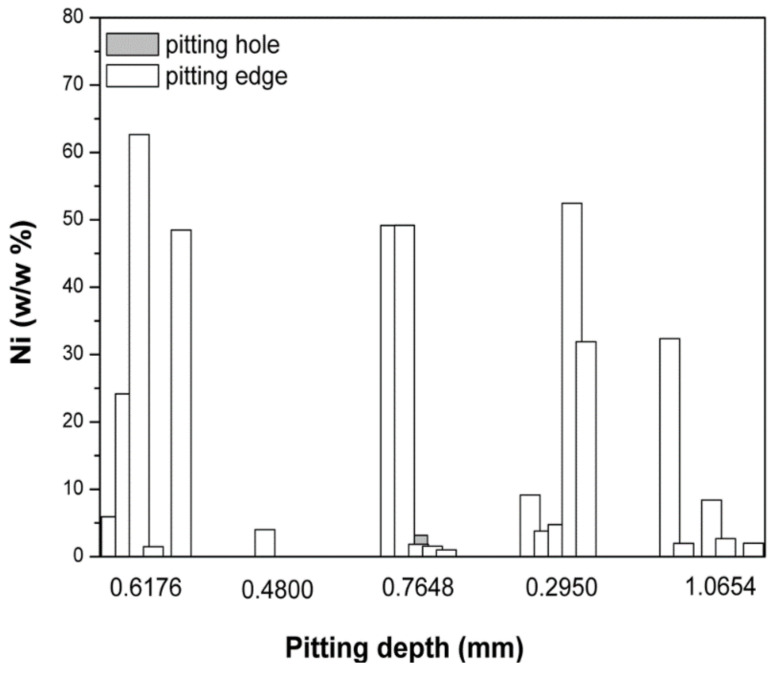
The content of Ni on the pit edges and inside the pits, based on the EDX analysis.

**Figure 9 micromachines-13-01031-f009:**
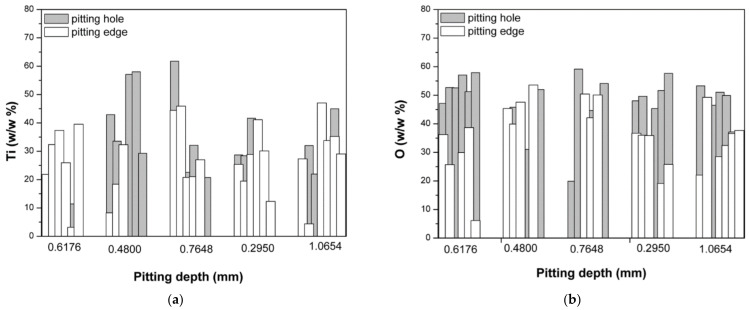
The contents of (**a**) Ti and (**b**) oxygen on the pit edges and inside the pits, based on the EDX analysis.

**Figure 10 micromachines-13-01031-f010:**
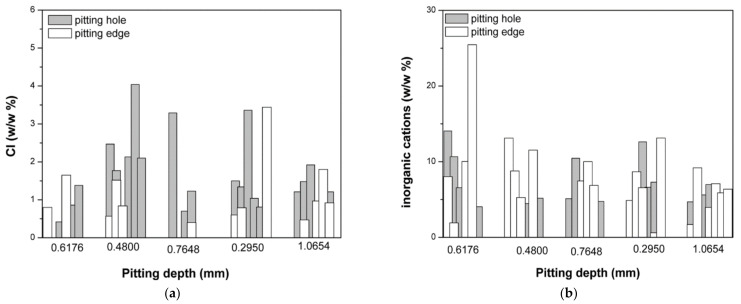
The content of (**a**) chloride and (**b**) inorganic cations on the pit edges and inside the pits, based on the EDX analysis.

**Table 1 micromachines-13-01031-t001:** Chemical composition of Ni and Ti in the initial test samples.

Element/Analyses	ICP (wt.%)	XRF (wt.%)
Ni	62.6	62.5–62.6
Ti	35.9	35.9

**Table 2 micromachines-13-01031-t002:** Pitting depth from SEM on the Ni–Ti surface and artificial measurements.

Measurement ^1^	SEM Pitting Depth		Artificial Pitting Depth Value			
	No.	Value	No.	Min.	Max.	Avg.
Measurement A	1	0.7787	20	0.074	0.778	0.544
Measurement B	1	0.4490	20	0.135	0.449	0.460
Measurement C	1	0.6322	20	0.107	0.632	0.413
Measurement D	1	1.3750	20	0.339	1.375	0.897
Measurement E	1	0.5493	20	0.186	0.549	0.423
Measurement F	1	0.6889	20	0.099	0.689	0.350

^1^ All values of pit depth are measured in mm.

**Table 3 micromachines-13-01031-t003:** Descriptive statistics of pitting corrosion measurements for the Ni–Ti sample.

Statistic	Value	Percentile	Value
Sample Size	120	Min	0.010
Range	0.181	5%	0.015
Mean	0.069	10%	0.017
Variance	0.002	First quartile	0.041
Standard Deviation	0.041	Median	0.067
Coefficient of Variation	0.587	Third quartile	0.088
Standard Error	0.004	90%	0.108
Skewness	1.036	95%	0.180
Excess Kurtosis	1.378	Max	0.192

**Table 4 micromachines-13-01031-t004:** Kolmogorov–Smirnov goodness of fit results.

**Statistic**	0.08428				
*p*-Value	0.3422				
α	0.2	0.1	0.05	0.02	0.01
Critical Value	0.09795	0.11164	0.12397	0.13857	0.14871
Reject?	No	No	No	No	No

**Table 5 micromachines-13-01031-t005:** EDX spectra for Figure 5b, representing Hole Site 5 from Figure 5a.

Spectrum	C	O	Na	Mg	Al	Si	S	Cl	K	Ca	Ti	Fe	Total
Spectrum 1	17.10	53.30	1.67	0.77	0.69	1.25		1.21		2.25	21.76		100.00
Spectrum 2	13.80	43.91	0.97	0.88	1.32	2.44		1.48		3.18	32.01		100.00
Spectrum 3	22.29	46.48	1.64	1.33	0.52	0.72	0.52	1.92		2.64	21.95		100.00
Spectrum 4	10.75	51.04	0.93	1.49	7.34	15.76			2.36	4.56	1.17	4.60	100.00
Spectrum 5	29.15	49.91	2.18	1.34				1.13		2.92	13.38		100.00
Spectrum 6	11.48	37.16			0.50	0.74		1.21		3.91	44.99		100.00
Max.	29.15	53.30	2.18	1.49	7.34	15.76	0.52	1.92	2.36	4.56	44.99	4.60	
Min.	10.75	37.16	0.93	0.77	0.50	0.72	0.52	1.13	2.36	2.25	1.17	4.60	

**Table 6 micromachines-13-01031-t006:** The results of EDX analyses of Ti and Ni on the surface of the Ni–Ti alloy.

	Pitting Edge (Total 30 Spectrums)			Pitting Hole (Total 30 Spectrums)		
	Min	Max	Mean	Min	Max	Mean
Ti	3.20	90.39	27.49	1.03	61.77	24.76
Ni	1.00	62.66	13.30	3.17	38.37	1.38
Sea Product Elements	/	/	Rest	/	/	Rest

All results in weight %.

**Table 7 micromachines-13-01031-t007:** The average chemical composition of the Ni–Ti surface based on the EDX analysis.

	Average Chemical Composition (wt. %)										
Month	O	Na	Mg	Si	S	Cl	K	Ca	Ti	Ni	C
6	29.84	4.11	1.29	0.35	0.19	0.76	0.038	0.19	29.25	33.27	0
12	28.50	1.59	0.50	2.65	0.80	3.17	0	1.05	13.31	14.26	33.17
18	37.28	1.20	1.48	7.03	0.83	0.95	1.71	2.11	13.72	13.22	15.10

## Data Availability

Not applicable.

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
