# Peer review of "A Nonlinear Probabilistic Pitting Corrosion Model of Ni–Ti Alloy Immersed in Shallow Seawater"

_micromachines, 2022, doi:10.3390/mi13071031_

Round 1
Reviewer 1 Report
The pitting corrosion model of Ni-Ti alloy was studied in this work and some interesting results have been obtained. The paper can be considered for publication after the following revisions:
1. For table 5, the wt% or at% should be marked in the EDS results for readers.
2. How about the precision on the Chloride ions content determined by EDS? Are there any comparable literatures? Could the authors add some related work for clear comparison?
3. In the abstract section, the authors stated that ICP and XRF were used. However, in the main body of the paper, only EDS results were given. Besides, no spectrum was given for the EDS results. Could the authors give the EDS spectrum data?
4. What is the objective of the fitted model? How does it work for prediction work or any other use, which should be clearly presented for readers?
Reviewer 2 Report
This manuscript presents the results of the long-therm corrosion investigations of NiTi alloy in seawater. From the practical point of view, this is very interesting and valuable research. It is a very good written manuscript. The methods are very well described, results are explained in detail, the conclusions are adequately derived.
However, I have a few suggestions for minor correction in the manuscript for the authors:
In Abstract section, line 26, authors has written that "the corrosion starts immediately after 12 months" - the word immediately should be omitted as because it would indicate that corrosion starts immediately after immersion in the sea, not after 12 months.
In 2.1. Preparation of the Ni-Ti Alloy; line 110/111 "... which was connected to a 60 kW medium frequency (4 kHz)" I think that something is missing in this sentence - connected to what?
Line 125 "(list the works Rijeka, Springer and the work Kristal Maritime Rivers) - this part of the sentence requires additional clarification.
Line 135 - rapid corrosion process; it is not rapid if it starts after 12 months
The references are written in different styles, it needs to be unified.
Please correct the reference 47 in this way:
Kožuh, S; Vrsalović, L.; Gojić, M.; Gudić, S.; Kosec, B. Comparison of the corrosion behavior and surface morphology of NiTi alloy and stainless steels in sodium chloride solution, J. Min. Metall. B, 2016, 52, 53-61.
Round 2
Reviewer 1 Report
It has been revised carefully and can be accepted now.